# Statistical Analysis of the Axillary Temperatures Measured by a Predictive Electronic Thermometer in Healthy Japanese Adults

**DOI:** 10.3390/ijerph18105096

**Published:** 2021-05-12

**Authors:** Tatsuya Yoshihara, Masayoshi Zaitsu, Kazuya Ito, Eunhee Chung, Mayumi Matsumoto, Junko Manabe, Takashi Sakamoto, Hiroshi Tsukikawa, Misato Nakagawa, Masami Shingu, Shunji Matsuki, Shin Irie

**Affiliations:** 1SOUSEIKAI Fukuoka Mirai Hospital Clinical Research Center, Kashiiteriha 3-5-1, Higashi-ku, Fukuoka 813-0017, Japan; mayumi-matsumoto@lta-med.com (M.M.); junko-manabe@lta-med.com (J.M.); takashi-sakamoto@lta-med.com (T.S.); hiroshi-tsukikawa@lta-med.com (H.T.); misato-nakagawa@lta-med.com (M.N.); masami-shingu@lta-med.com (M.S.); shunji-matsuki@lta-med.com (S.M.); shin-irie@lta-med.com (S.I.); 2Department of Public Health, Dokkyo Medical University School of Medicine, 880 Kitakobayashi, Mibu-machi, Shimotsuga-gun, Tochigi 321-0293, Japan; m-zaitsu@dokkyomed.ac.jp; 3SOUSEIKAI Clinical Epidemiological Research Center, Kashiiteriha 3-5-1, Higashi-ku, Fukuoka 813-0017, Japan; kazuya-ito@lta-med.com; 4College of Healthcare Management, Takayanagi 960-4, Setaka-machi, Miyama 835-0018, Japan; 5SOUSEIKAI Global Clinical Research Center, Kashiiteriha 3-5-1, Higashi-ku, Fukuoka 813-0017, Japan; eunhee-chung@lta-med.com

**Keywords:** body temperature, axillary temperature, distribution, healthy Japanese adults, predictive electronic thermometer

## Abstract

Body temperature is important for diagnosing illnesses. However, its assessment is often a difficult task, considering the large individual differences. Although 37 °C has been the gold standard of body temperature for over a century, the temperature of modern people is reportedly decreasing year by year. However, a mean axillary temperature of 36.89 ± 0.34 °C reported in 1957 is still cited in Japan. To assess the measured axillary temperature appropriately, understanding its distribution in modern people is important. This study retrospectively analyzed 2454 axillary temperature measurement data of healthy Japanese adults in 2019 (age range, 20–79 years; 2258 males). Their mean temperature was 36.47 ± 0.28 °C (36.48 ± 0.27 °C in males and 36.35 ± 0.31 °C in females). Approximately 5% of the 20–39-year-old males had body temperature ≥37 °C, whereas 8% had a temperature ≥ 37 °C in the afternoon. However, none of the subjects aged ≥50 years reported body temperature ≥37 °C. In multivariable regression analysis, age, blood pressure, pulse rate, and measurement time of the day were associated with axillary temperature. Our data showed that the body temperature of modern Japanese adults was lower than that reported previously. When assessing body temperature, the age, blood pressure, pulse rate, and measurement time of the day should be considered.

## 1. Introduction

Body temperature is one of the key vital signs that provide information about a patient’s physiological condition in a non-invasive and simple manner. Thus, it is important in the health management, diagnosis of various illnesses including infectious diseases such as coronavirus disease (COVID-19), and evaluation of the severity of the illness. However, there is great individual variability in body temperature. A recent large cohort study reported great individual differences in normal body temperature, with mean body temperature of 36.6 °C (95% range 35.7–37.3 °C, 99% range 35.3–37.7 °C) [1]. Therefore, its assessment is often a difficult task. To make an appropriate assessment of the measured body temperature, it is necessary to understand the body temperature distribution among healthy individuals.

In 1868, the average axillary temperature in a German population was reported to be 37 °C, which has been the gold standard of body temperature for more than a century [2]. However, in 2020, Protsiv et al. reported that the body temperature of modern humans had been decreasing by 0.03 °C every 10 years [3]. This could be attributed to a decrease in physical activities, changes in lifestyle habits such as reduced physical activities, the use of air conditioning, reduction of chronic infections, or use of non-steroidal anti-inflammatory drugs (e.g., paracetamol or metamizole) [3].

In clinical practice, the body temperature can be measured at various sites, such as the mouth, axilla, rectum, or ear. The body temperature is widely measured orally in adults in Western countries [1,4], because it is the preferred method of obtaining an accurate body temperature [5]. On the other hand, axillary temperature measurement is the common method used in children and unconscious patients because of the ease of access [4]. In Japan, axillary temperature measurement has been widely used in both children and adults not only at hospitals, but also at home. However, the axillary temperature has been reported to be lower than the oral temperature by 0.1–0.5 °C [4,5,6,7,8]. Furthermore, the measurement methods have changed from the use of mercury thermometers to electronic thermometers in the past few decades. Considering the decline in body temperature in modern humans, the difference between the oral and axillary temperatures, and the changes in the measurement instruments, it is essential to understand the updated axillary temperature distribution measured using predictive electronic thermometers in healthy adults for its appropriate assessment in clinical practice. However, currently, recent reliable large-scale data from various age groups are not available [9]. Therefore, a mean temperature of 36.89 ± 0.34 °C, which was reported in 1957 [10], is still cited in Japan as the benchmark of axillary temperature in healthy Japanese adults [11].

This study analyzed the axillary temperature data from 2027 of healthy Japanese adults, who had undergone physical examinations for participating in clinical trials at our clinical research center. Since body temperature has been reported to be affected by various factors, such as age, body mass index (BMI), measurement time of the day, or season [1,2,3,12,13], the effects of these factors on body temperature were also investigated.

## 2. Materials and Methods

### 2.1. Ethics

The study was approved by the SOUSEIKAI Hakata Clinic Clinical Trials Review Board (Approval number: N-69). It was conducted in accordance with the Declaration of Helsinki and the Ethical Guidelines for Medical and Health Research Involving Human Subjects in Japan.

### 2.2. Research Subjects

We retrospectively collected and analyzed the anonymous data of 2454 (2258 men and 196 women) axillary temperature measurements from 2027 (1835 men and 192 women) participants aged 20 to 79 years, who were not under medical treatment, were not using any drugs, and did not have any symptoms to suspect illness. These patients had undergone pre-trial health examination for clinical trials (mainly phase 1 trials) at the SOUSEIKAI Fukuoka Mirai Hospital Clinical Research Center from January 2019 to December 2019. We were not able to assess the laboratory data and were not able to exclude participants with underlying inflammatory diseases because anonymous data of body temperature were collected.

### 2.3. Measurement Method of Body Temperature

Body temperature was measured in the axilla using an electronic thermometer C231 (Terumo, Tokyo, Japan) that displays the temperature in increments of 0.1 °C. According to the product documentation of the thermometer C231, the maximum acceptable error is within 0.1 °C, and the measurable range is 32.0–42.0 °C [11]. The device required an average of 20 s to measure the predictive temperature, which was calculated using an arithmetic expression obtained from the processed data. The device can predict the 10 min equilibrium axillary temperature to reduce measurement time. Since the physical examination was conducted after obtaining informed consent for clinical trials, the body temperature was measured after sitting for at least 30 min to listen to the trial information. Moreover, the data used in this study was the body temperature that was measured on an empty stomach, because the blood tests in these clinical trials had to be conducted under fasting conditions.

### 2.4. Statistical Analysis

The body temperature data were indicated as mean ± standard deviation, up to two decimal places. Body temperatures were measured in the morning (from 9:00 to 11:59, hereafter referred as “A.M.”) for 1522 participants and in the afternoon (from 13:00 to 15:59, hereafter referred as “P.M.”) for 932 participants. To investigate the effects of various parameters on body temperature, including the age, sex, BMI, systolic blood pressure (SBP), diastolic blood pressure (DBP), pulse rate, and measurement time of the day, we used an ordinary least squares regression model to estimate the regression coefficients and 95% confidence intervals in a multivariable linear regression analysis. JMP Pro 15 (SAS Institute Inc., Tokyo, Japan) was used for all analyses.

## 3. Results

The overall mean axillary temperature of the participants was 36.47 ± 0.28 °C (Table 1); however, the distribution was slightly skewed to the left (the coefficients of skewness were −0.17 in males and −0.43 in females) (Figure 1). There was a difference in the mean body temperature (male, 36.48 ± 0.27 °C; female, 36.35 ± 0.31 °C; *p* < 0.001) and mean age between the males and females (male, 27.5 ± 8.9 years; female, 47.6 ± 15.9 years; *p* < 0.001) (Table 1). The mean body temperatures in P.M. were higher than those in A.M. (Table 1).

Linear regression analyses seen in Figure 2A,B showed that body temperature was correlated with age in males and females (R = 0.17, β (95% confidence interval [CI]) = −0.005 (−0.006 to −0.004), Y-intercept = 36.62, *p* < 0.001 in males, R = 0.41, β (95% CI) = −0.008 (−0.01 to −0.005), Y-intercept = 36.72, *p* < 0.001 in females).

In the young population, 4.6–4.7% of the males aged 20–39 years and 2.0% of the females aged 20–29 years had a body temperature of 37 °C or higher (Table 2). In the P.M. measurements, 7.4–8.1% of the males aged 20–39 years reported a temperature of 37 °C or higher. In contrast, none of the subjects who were aged 50 years or older reported a temperature of 37 °C or higher, although there were no statistically significant differences among the age groups (Table 2).

The mean body temperature according to different seasons was as follows: 36.47 ± 0.28 °C in winter, 36.50 ± 0.28 °C in spring, 36.45 ± 0.29 °C in summer, and 36.45 ± 0.26 °C in autumn. In the univariable regression analysis, age, female sex, and DBP were negatively associated with body temperature, whereas pulse rate, spring, and P.M. measurement were positively associated (Table 3). The multivariable regression analysis showed that age, SBP, DBP, pulse rate, and measurement time (A.M. or P.M.) had a significant effect on body temperature (Table 3).

## 4. Discussion

In this study, the data from 2454 body temperature measurements of 2027 healthy Japanese adults measured in 2019 were analyzed. The mean body temperature was 36.47 ± 0.28 °C, showing a negative association with age and DBP and a positive association with SBP, pulse rate, and P.M. measurement. The factors of sex, BMI, and seasons had no influence on body temperature.

The body temperature of modern humans is reportedly decreasing. In 1868, Wunderlich reported a mean axillary temperature of 37 °C [3]. In 1992, Mackowiak et al. reported a mean temperature of 36.8 °C measured orally [2]. In 2017, Obermeyer et al. reported a mean temperature of 36.6 °C measured orally based on the data of 243,506 measurements obtained from 35,488 patients in the United States [1]. In Japan, Tasaka et al. reported a mean axillary temperature of 36.89 ± 0.34 °C in 3094 healthy volunteers (approximately 10–50 years of age) in 1957 [10], which is still cited as the representative axillary temperature of Japanese people [11]. In 1988, Iriki et al. reported a mean axillary temperature of 36.72 ± 0.36 °C in healthy Japanese adults [6]. The mean axillary temperature of 36.47 °C in this study is 0.2–0.4 °C lower than that reported in the aforementioned studies. This also might indicate a slight decrease in the body temperature of humans in recent years.

Oral measurements have been widely adopted in Europe and the United States, while axillary measurements are commonly used in Japan. The reasons why oral temperature measurements have not been widely accepted in Japan remain unclear. However, the ease of access, risk of oral trauma caused by the glass breakage of mercury thermometers, and fear of the side effects of mercury could be the contributing factors [6,9]. Although mercury thermometers are no longer used in Japan as well as in Europe and the United States, and electronic oral thermometers and ear thermometers are commercially available, axillary temperature measurements are still the most common measurement method in Japan. Although the axillary temperature is reported to be closely associated with the core temperature [9], the axillary temperature has been criticized for being only a modified external temperature and susceptible to environmental temperature, hypotension, or skin vasodilation [6,14]. The mean body temperature of 36.47 °C in this study was approximately 0.1 °C lower than the oral temperature of 36.6 °C reported in the United States in 2017 [1], which might support the previous reports that axillary temperature is slightly lower than the oral temperature by 0.1–0.5 °C [4,5,6,7,8].

In this study, body temperature was found to be negatively associated with age. The results of the multivariable regression analysis indicated that body temperature decreased by 0.005 °C with every one-year increase in age. It has been reported that older people have lower body temperatures than younger people [1,2,15]. Obermeyer et al. reported a 0.021 °C decrease in oral temperature for every 10 years of age [1]. The influence of age on body temperature in the report by Obermeyer was lesser than that seen in the present study (0.05 °C decrease every 10 years of age). The reason for this difference could be the measurement methods, because older people’s skin temperature tends to be lower than that of younger people [16]. Therefore, further investigations are needed to clarify the differences between axillary and oral temperatures in older populations.

Body temperature is known to show circadian changes within 0.5–1 °C, being lowest early in the morning and highest in the afternoon to evening [1,2,12,13]. This is consistent with the results of this study. In this study, SBP, DBP, and pulse rate were associated with body temperature. It has been reported that pulse rate and DBP are positively associated with body temperature and are considered to be related to sympathetic activation [1]. We empirically observed that blood pressure and pulse rate often increase in patients with fever. Blood pressure and pulse rate should also be taken into account when determining whether a person has fever.

Body temperature has been reported to be positively associated with BMI [1,17]. The reason for this could be that subcutaneous fat keeps heat, and high calorie intake or more skeletal muscles generate a lot of heat [1]. However, BMI had little effect on body temperature in this study. The mean BMI in this study was 21.7 ± 2.0 kg/m^2^, and most values were in the normal range; therefore, the effect of BMI on body temperature might not have been elicited.

Sex differences in body temperature are controversial [18]. In this study, although the univariable regression analysis showed sex differences in body temperature, the multivariable regression analysis showed no sex differences. This may be due to the higher mean age of women than men. However, the ovulation cycle in females that is known to affect the body temperature was not considered. Further investigations that consider the ovulation cycle should be conducted for accurate comparisons of the sex differences.

Approximately 5% of the young males aged 20–39 years had an axillary temperature of 37 °C or higher, whereas 8% of these males had a body temperature of 37 °C or higher in the P.M. measurements. In contrast, none of the subjects who were over 50 years of age reported a temperature of 37 °C or higher, although there were no statistically significant differences among the age groups. A body temperature of 37 °C or higher is occasionally assessed as an indicator of fever. However, young people often have a temperature above 37 °C, especially in the afternoon, as shown in our study. Thus, the measured body temperature should be assessed taking into consideration the body temperature distribution depending on the person’s age.

### Strengths and Limitations of the Study

Body temperature is sensitive to exercise, diet, axillary sweating, environmental temperature, and drug use [3]. At our clinical research center, the body temperature was measured after the subjects were seated in fasting condition for at least 30 min at a room temperature of approximately 25 °C. All subjects confirmed that they had not received any medical treatment at the time of temperature measurement. Thus, the effects of exercise, diet, axillary sweating, environmental temperature, and drug use were eliminated in this study. Seasonal effects on body temperature have also been reported [1]. However, our data did not show any apparent seasonal effects. This may be because body temperatures were measured in similar environmental conditions in all the seasons. The year of measurement or presence of COVID-19 did not affect the results of this study, because we used body temperature data that was measured only in 2019.

This study had some limitations. First, the axillary temperatures were measured from the unilateral axilla in each subject, though it has been reported that the average temperature from the bilateral axillae is more accurate than the unilateral measurement [4]. Second, the number of older people and females was relatively small, because the data were from the clinical trials conducted in 2019; this may be a source of bias. Further large-scale studies are needed to reveal the accurate distribution of older people and women. Third, it cannot be completely ruled out that the participants had undiagnosed acute illnesses, such as infectious diseases, at the time of the body temperature measurement. We were not able to collect the laboratory data of the subjects, such as white blood cell counts or C-reactive protein, because the body temperature data were kept anonymous. However, the subjects’ health conditions in this study were considered good, because we always asked them to refrain from coming for the physical examination when they had any health problems or were feeling unwell. Additionally, the health condition was always double-checked at the time of their visit. Fourth, this study included multiple data points in some subjects because they participated in clinical trials more than once a year, which might be a source of bias. However, the number of subjects was relatively small (316 out of 2027), their body temperatures were measured on separate days, and the previous studies also included multiple measurements in the same subject [1,3,8], therefore we think that the influence of the bias might be limited. Fifth, the variables in this study were selected in accordance with previous studies [1,3], however, the number of variables was limited due to the nature of the data that we had. Therefore, large-scale studies with robust analysis [19,20] involving many variables are needed in the future to reveal the factors that influence body temperature.

## 5. Conclusions

In this study, the mean axillary temperature of healthy young Japanese adults was approximately 36.5 °C, which was lower than that reported in previous studies in Japan. The axillary temperature decreased with increasing age. Moreover, it was found to be higher when measured in the afternoon than in the morning. Additionally, blood pressure and pulse rate were associated with body temperature. Since certain young people may have a temperature of 37 °C or higher, the age, time of measurement, blood pressure, and pulse rate must be taken into account when assessing the body temperature.

## Figures and Tables

**Figure 1 ijerph-18-05096-f001:**
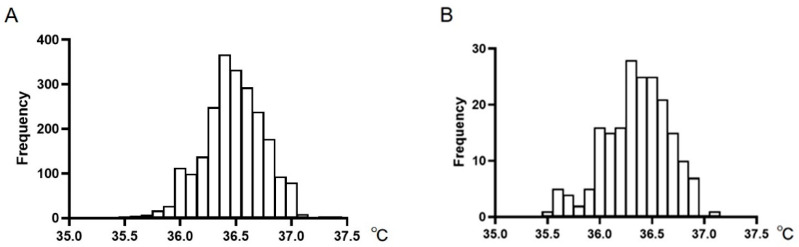
Distribution of the axillary temperature measurements in healthy Japanese male adults (*n* = 2258) (**A**) and female adults (*n* = 196) (**B**).

**Figure 2 ijerph-18-05096-f002:**
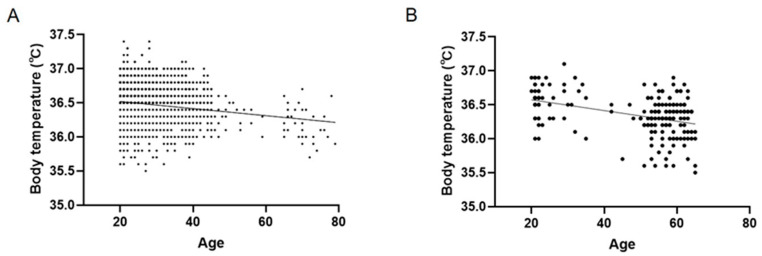
Correlation of age and body temperature in healthy Japanese male adults (**A**) and female adults (**B**).

**Table 1 ijerph-18-05096-t001:** Characteristics of the body temperature measurement.

Characteristics	All	Male	Female	*p*-Value ^1^
*n*	2454	2258	196	
Mean age (SD), years	29.1 (11.1)	27.5 (8.9)	47.6 (15.9)	<0.001
Age range, years	20–79	20–79	20–65	
Mean body temperature (SD), °C	36.47 (0.28)	36.48 (0.27)	36.35 (0.31)	<0.001
Body temperature range, °C	35.5–37.4	35.5–37.4	35.5–37.1	
Mean body temperature in A.M. (SD), °C	36.42 (0.27)	36.44 (0.26)	36.29 (0.30)	
Mean body temperature in P.M. (SD), °C	36.54 (0.28)	36.54 (0.28)	36.55 (0.26)	
*p*-value ^2^	<0.001	<0.001	<0.001	

^1^ Differences between the male and female groups were analyzed by the *t*-test. ^2^ Differences between the mean body temperatures measured in the morning (9:00–11:59, indicated as “A.M.”) and in the afternoon (13:00–15:59, indicated as “P.M.”) were analyzed by the *t*-test. SD, standard deviation; A.M., measurements in the morning; P.M., measurements in the afternoon.

**Table 2 ijerph-18-05096-t002:** Mean body temperature stratified by age and sex and the ratio of the subjects whose body temperature was 37 °C or higher.

Age Group (Years)	Male				Female			
	*n*	mBT (SD) °C	≥37 °C % ^1^	≥37 °C (P.M.) % ^2^	*n*	mBT (SD) °C	≥37 °C % ^1^	≥37 °C (P.M.) % ^2^
20–29	1595	36.50 (0.27)	4.6	8.1	49	36.56 (0.25)	2.0	2.3
30–39	490	36.46 (0.28)	4.7	7.4	8	36.51 (0.32)	0	0
40–49	123	36.43 (0.24)	1.6	1.7	5	36.26 (0.38)	0	-
50–59	12	36.30 (0.15)	0	0	88	36.28 (0.29)	0	-
60–69	13	36.28 (0.20)	0	0	46	36.27 (0.30)	0	-
70–79	25	36.15 (0.25)	0	0	-	-	-	-
*p* value ^3^			0.64	0.55			0.55	0.92

^1^ The ratio of the subjects whose axillary temperature was 37 °C or higher. ^2^ The ratio of the subjects whose axillary temperature was measured in the afternoon (13:00–15:59, indicated as “P.M.”) and was 37 °C or higher; mBT, mean body temperature. ^3^ Fisher’s exact test; SD, standard deviation; P.M., measurements in the afternoon.

**Table 3 ijerph-18-05096-t003:** Regression coefficients of background parameters for body temperature estimated using ordinary least squares regression models.

Characteristics	Regression Coefficient (95% Confidence Interval)			
	Univariable Linear Regression	*p*-Value	Multivariable Linear Regression	*p*-Value
Age (years)	−0.006 (−0.007 to −0.005)	<0.001	−0.005 (−0.007 to −0.004)	<0.001
Sex (female)	−0.06 (−0.08 to −0.04)	<0.001	−0.003 (−0.03 to 0.02)	0.78
BMI (kg/m^2^)	−0.003 (−0.009 to 0.002)	0.22	0.005 (−0.0005 to 0.01)	0.08
SBP (mmHg)	−2.8 × 10^−5^ (−0.001 to 0.001)	0.96	0.002 (0.0003 to 0.003)	0.01
DBP (mmHg)	−0.003 (−0.004 to −0.001)	<0.001	−0.003 (−0.004 to −0.0009)	0.003
Pulse rate (beats per min)	0.006 (0.005 to 0.007)	<0.001	0.005 (0.004 to 0.006)	<0.001
Seasons				
Winter	Reference		Reference	
Spring	0.03 (0.01 to 0.05)	0.002	0.01 (−0.007 to 0.03)	0.23
Summer	−0.02 (−0.03 to 0.003)	0.10	0.01 (−0.005 to 0.03)	0.15
Autumn	−0.02 (−0.04 to 0.001)	0.07	−0.007 (−0.03 to 0.01)	0.44
Measurement time (P.M.)	0.06 (0.05 to 0.07)	<0.001	0.04 (0.03 to 0.05)	<0.001
adjusted R^2^			0.13	

BMI, body mass index; SBP, systolic blood pressure; DBP, diastolic blood pressure; R^2^, coefficients of determination; P.M., the afternoon measurement time of body temperature (13:00–15:59).

## Data Availability

The data that support the findings of this study are available from the corresponding author, TY, upon reasonable request, according to the Ethical Guidelines for Medical and Health Research Involving Human Subjects, Japan.

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
