# Peer review of "Statistical Analysis of the Axillary Temperatures Measured by a Predictive Electronic Thermometer in Healthy Japanese Adults"

_ijerph, 2021, doi:10.3390/ijerph18105096_

Round 1

Reviewer 1 Report

An interesting physiological assessment of temperature.  Overall well presented.  A few areas could be firmed up a little to help demonstrate clinical relevance. Also there are some areas that need to be presented with greater clarity. details are below, but in general, it would be good to identify resolution / accuracy of clinically used thermometers, as the differences identified may be outside the accuracy thresholds of some equipment.  this would also help establish the clinical relevance of your findings. 

There needs to be more clarity regarding commonly used thermometers outside of japan, as the manner in which this issue has been dealt with in the manuscript is not accurate (see detail below).  

owing to the disparity in sample sizes It would be good to separate the male and female temperatures clearly in the abstract (or remove female data and just state male).

Although identified in limitations, use of a more generic inflammatory marker such as C reactive protein may be a better than WBC counts.  

Page 1 Line 26.  Two subgroups have been identified with percentages given, were these groups statistically different from the main group.

Page 1 line 41 reference has been made to great individual variability in ain body temperature, the range of normal ideally supported by references would be useful to establish this argument.

Page 2 Line 47 according to your stated statistics the estimated drop in temperature is 0.3 degrees centigrade in the last 100 years, to help justify your perspective, it would be good to see what is the accuracy/ resolution of a thermometer in clinical use.

Page 2 line 52. I would go for a more contemporary reference than (3) as this was published in 2000, since then the use of other forms of thermometer (tympanic and thermodots) has been widespread, is oral really still the main approach used? If so a more contemporary reference is needed to establish this point.

Page 5 line 160-165 The EU and US have pretty much band oral glass thermometers (mercury) see https://www.legislation.gov.uk/eudr/2007/51 for the last 20 odd years. 

Page 5 line 173 So your study agrees with the fact that axillary temperature is lower than oral temperature as stated in line 171?

Page 6 202 -203. Were these raises statistically significant compared to the rest of the group

Author Response

Response to Reviewer 1 Comments

An interesting physiological assessment of temperature.  Overall well presented.  A few areas could be firmed up a little to help demonstrate clinical relevance. Also there are some areas that need to be presented with greater clarity. details are below, but in general, it would be good to identify resolution / accuracy of clinically used thermometers, as the differences identified may be outside the accuracy thresholds of some equipment.  this would also help establish the clinical relevance of your findings. 

There needs to be more clarity regarding commonly used thermometers outside of japan, as the manner in which this issue has been dealt with in the manuscript is not accurate (see detail below).  

We thank the reviewer for these comments. Our point-by-point responses to each of the reviewer’s comments are as follows:

Point 1: owing to the disparity in sample sizes It would be good to separate the male and female temperatures clearly in the abstract (or remove female data and just state male).

Response 1: We thank the reviewer for this comment. We separated the male and female temperature in the abstract as follows:

Abstract, Line 26:

(Original): Their mean temperature was 36.47 ± 0.28 °C.

(Revised, revised parts are highlighted in yellow): Their mean temperature was 36.47 ± 0.28 °C (36.48 ± 0.27 °C in males and 36.35 ± 0.31 °C in females).

Also, in accordance with the comment of reviewer 4, we replaced the histogram (Figure 1) for separating the male and female temperature. Please check the revised Figure 1A and 1B in the revised manuscript.

Point 2: Although identified in limitations, use of a more generic inflammatory marker such as C reactive protein may be a better than WBC counts.  

Response 2: We thank the reviewer for this comment. We changed the sentences as follows:

Discussion, Line 257-261:

(Original):

Third, although it has been reported that higher white blood cell counts are associated with higher temperatures [17], we were not able to collect the laboratory data of the subjects because the body temperature data were kept anonymous. Fourth, it cannot be completely ruled out that the participants had undiagnosed acute illnesses, such as infectious diseases, at the time of the body temperature measurement.

(Revised, revised parts are highlighted in yellow):

Third, it cannot be completely ruled out that the participants had undiagnosed acute illnesses, such as infectious diseases, at the time of the body temperature measurement. We were not able to collect the laboratory data of the subjects, such as white blood cell counts or C-reactive protein, because the body temperature data were kept anonymous. 

Point 3: Page 1 Line 26.  Two subgroups have been identified with percentages given, were these groups statistically different from the main group.

Response 3: We thank the reviewer for this comment. We added the results of statistical analyses among the age groups in Table 2. Please check the revised Table 2 in the revised manuscript.

Also, we changed the text in the Results section as follows:

Results, Line 137-142:

(Original):

In the young population, 4.6–4.7% of the males aged 20–39 years and 2.0% of the females aged 20–29 years had a body temperature of 37 °C or higher (Table 2). In the P.M. measurements, 7.4–8.1% of the males aged 20–39 years reported a temperature of 37 °C or higher. However, none of the subjects who were aged 50 years or older reported a temperature of 37 °C or higher.

(Revised, revised parts are highlighted in yellow):

In the young population, 4.6–4.7% of the males aged 20–39 years and 2.0% of the females aged 20–29 years had a body temperature of 37 °C or higher (Table 2). In the P.M. measurements, 7.4–8.1% of the males aged 20–39 years reported a temperature of 37 °C or higher. In contrast, none of the subjects who were aged 50 years or older reported a temperature of 37 °C or higher, although there were no statistically significant differences among the age groups (Table 2).

Point 4: Page 1 line 41 reference has been made to great individual variability in ain body temperature, the range of normal ideally supported by references would be useful to establish this argument.

Response 4: We thank the reviewer for this comment. We added the range of normal in a previous large cohort study [1] as follows.

Introduction, Line 42-45:

(Original):

However, because there is great individual variability in body temperature, its assessment is often a difficult task.

(Revised, revised parts are highlighted in yellow):

However, there is great individual variability in body temperature. A recent large cohort study reported great individual differences in normal body temperature, with mean body temperature of 36.6 °C (95% range 35.7–37.3 °C, 99% range 35.3–37.7 °C) [1]. Therefore, its assessment is often a difficult task.

Point 5: Page 2 Line 47 according to your stated statistics the estimated drop in temperature is 0.3 degrees centigrade in the last 100 years, to help justify your perspective, it would be good to see what is the accuracy/ resolution of a thermometer in clinical use.

Response 5: We thank the reviewer for this comment. We added a sentence that explains the accuracy of the thermometer used in this study.

Materials and Methods, Line 96-98:

(Original):

Body temperature was measured in the axilla using an electronic thermometer C231 (Terumo, Tokyo, Japan) that displays the temperature in increments of 0.1 °C.

(Revised, revised parts are highlighted in yellow):

Body temperature was measured in the axilla using an electronic thermometer C231 (Terumo, Tokyo, Japan) that displays the temperature in increments of 0.1 °C. According to the product documentation of the thermometer C231, the maximum acceptable error is within 0.1 °C, and the measurable range is 32.0-42.0 °C [11].

Point 6: Page 2 line 52. I would go for a more contemporary reference than (3) as this was published in 2000, since then the use of other forms of thermometer (tympanic and thermodots) has been widespread, is oral really still the main approach used? If so a more contemporary reference is needed to establish this point.

Response 6: We thank the reviewer for this comment. We think that oral temperature is most commonly measured in the Unites States, as described by Obermeyer et al. (reference [1] published in 2017). However, we do not know much about Europe. Therefore, we added a reference [1] and changed the sentence as follows:

Introduction, Line 56-58:

(Original):

The body temperature is most commonly measured orally in adults in Western countries [3], because it is the preferred method of obtaining an accurate body temperature [4].

(Revised, revised parts are highlighted in yellow):

The body temperature is widely measured orally in adults in Western countries [1,4], because it is the preferred method of obtaining an accurate body temperature [5].

Point 7: Page 5 line 160-165 The EU and US have pretty much band oral glass thermometers (mercury) see https://www.legislation.gov.uk/eudr/2007/51 for the last 20 odd years. 

Response 7: We thank the reviewer for this comment. We changed the sentence as follows:

Discussion, Line 187-191:

(Original):

Although mercury thermometers are no longer used in Japan and electronic oral thermometers and ear thermometers are commercially available, axillary temperature measurements are still the most common measurement method in Japan.

(Revised, revised parts are highlighted in yellow):

Although mercury thermometers are no longer used in Japan as well as in Europe and the United States, and electronic oral thermometers and ear thermometers are commercially available, axillary temperature measurements are still the most common measurement method in Japan.

Point 8: Page 5 line 173 So your study agrees with the fact that axillary temperature is lower than oral temperature as stated in line 171?

Response 8: We thank the reviewer for this comment. We changed the sentences as follows:

Discussion, Line 194-198:

(Original):

In fact, axillary temperature was reported to be lower than the oral temperature by 0.1–0.5 °C [3–7]. The mean body temperature of 36.47 °C in this study was approximately 0.1 °C lower than the oral temperature of 36.6 °C reported in the United States in 2017 [13].

(Revised, revised parts are highlighted in yellow):

The mean body temperature of 36.47 °C in this study was approximately 0.1 °C lower than the oral temperature of 36.6 °C reported in the United States in 2017 [1], which might support the previous reports that axillary temperature is slightly lower than the oral temperature by 0.1–0.5 °C [4–8]..

Point 9: Page 6 202 -203. Were these raises statistically significant compared to the rest of the group

Response 9: We thank the reviewer for this comment. We added the results of statistical analyses among the age groups in Table 2. Please check the revised Table 2 in the revised manuscript.

Also, we changed the text in the Discussion section as follows:

Discussion, Line 230-234:

(Original):

Approximately 5% of the young males aged 20–39 years had an axillary temperature of 37 °C or higher, whereas 8% of these males had a body temperature of 37 °C or higher in the P.M. measurements. In contrast, none of the subjects who were over 50 years of age reported a temperature of 37 °C or higher.

(Revised, revised parts are highlighted in yellow):

Approximately 5% of the young males aged 20–39 years had an axillary temperature of 37 °C or higher, whereas 8% of these males had a body temperature of 37 °C or higher in the P.M. measurements. In contrast, none of the subjects who were over 50 years of age reported a temperature of 37 °C or higher, although there were no statistically significant differences among the age groups.

Reviewer 2 Report

The proposed study aims at conducting statistical analyses on axillary temperatures in healthy Japanese people. The major conclusions have been drawn based on results from linear regression using temperature as the response variable and relevant clinical/environmental factors as predictors. 

I agree with the authors that understanding the distribution of axillary temperatures in modern people is important. However, Figure 1 shows that the distribution in the sample under study is skewed and does not satisfy the normal distribution assumption for linear regression (you can also check whether the residuals are normal). Therefore, justifications are necessary to show the results in Table 3 are still valid.  

The covariates age, SBP, DBP, etc. have been shown to have significant effects on the response. It looks to me that this part of the analysis is related to variable selection. Given the doubt on the distributional assumption of the model, I suggested the author also include robust analyses in the study (Wu and Ma PMID: 25479793, Ren et al. PMID: 30746793) to support the results. For example, the R package regnet (based on Ren et al) can be directly applied on the data to select important predictors associated with the skewed response.  Are the results from robust analysis consistent with current ones?

The heavy-tailed distributions in the outcome variables have been widely encountered in statistical analysis. Please discuss the importance of robust statistical analysis and how it affects the proposed study.

Reviewer 3 Report

IJERPH
the decrease in body temperature with age does not surprise. but the decrease of body temperature with progress in civilization can only be explained by climiate changens. the environment is warming up and body does not need to produce heat. so there follows less physical activity and more weight gain. this is the chain of cause and consequence in modern civilization. 
line 50 and later so: the nonsteroidals such as ibuprofen have no antipyretic activity - but paracetamol or metamizole do so. 
line 88: what is a "predictive" temperature ... 
line 106: for me, the distribution in figure 1 looks skewed to the right
lines 119-121: formatting problem 
lines 121-126: a figure for the correlation analysis with a regression line an intercept and a slope and a correlation coefficient and a p - value would be more informative than the naked table on age and temperature. 
line 132: "SBP, DBP" means probably blood pressure but this is not explained
line 144-147: this is repetitive and could be cancelled
line 157-157: repetition - cancel
l.ine 170-172: again repetitive ... could be cancelled

wed 14 apr 2021

Reviewer 4 Report

Review for ijerph-1165991-peer-review-v1

In this paper, authors report a range for normal axillary temperature value for Japanese male and female populations.

The article is very well written and clear in describing the research hypothesis and findings.

Minor comments:

In the Methods, please add the following information: What were the inclusion and exclusion criteria? Were the subjects selected from a healthy group? If not, was there any underlying factor such as chronic inflammation in the subjects that could have affected their body temperature? Were the subjects taking any medications such as statins, Aspirin, or NSAIDs that could have affected their body temperature?  Was there any form of blood test done on subjects to verify if there was any underlying inflammation?

What were the accuracy and precision associated with the thermometer used in this study?

It seems that more than one data point was collected from some of the subjects. Would this be a source of bias? Please discuss.

The number of male participants in this study was an order of magnitude greater than that of female participants. Please discuss if this could have resulted in any form of bias as the female measurement results could have been masked because of their relatively much smaller number of data points.

On line 107, authors report a significant difference between male and female subjects in their measured body temperature. They also report a significant difference in age distribution between the two sexes. Could the difference in body temperature between male and female be, as a whole or partially, explained by participating female subjects being older on average. This possibility is further implied in the result of multivariable regression. To prevent confusion, please discuss the result of univariable analysis with consideration of the result from multivariable analysis where the effect of age and/or other confounding factors are also considered.

I suggest replacing Figure 1 with two figures one showing the histogram of temperature distribution for men and the other for women.

Authors state that the age should be taken into consideration when assessing body temperature, in that light it would be helpful if the authors could add another figure indicating average normal body temperature and its standard deviation for each decade of life.

Round 2

Reviewer 2 Report

I thank the reviewers for addressing my comments.